# TextGrad: Advancing Robustness Evaluation in NLP by Gradient-Driven Optimization

**Bairu Hou**[1], **Jinghan Jia**[2,*], **Yihua Zhang**[2,*], **Guanhua Zhang**[1,*],
**Yang Zhang**[3], **Sijia Liu**[2,3], **Shiyu Chang**[1]
[1]UC Santa Barbara, [2]Michigan State University, [3]MIT-IBM Watson AI Lab

## Abstract

Robustness evaluation against adversarial examples has become increasingly important to unveil the trustworthiness of the prevailing deep models in natural language processing (NLP). However, in contrast to the computer vision (CV) domain where the first-order projected gradient descent (PGD) is used as the benchmark approach to generate adversarial examples for robustness evaluation, there lacks a principled first-order gradient-based robustness evaluation framework in NLP. The emerging optimization challenges lie in 1) the discrete nature of textual inputs together with the strong coupling between the perturbation location and the actual content, and 2) the additional constraint that the perturbed text should be fluent and achieve a low perplexity under a language model. These challenges make the development of PGD-like NLP attacks difficult. To bridge the gap, we propose TextGrad, a new attack generator using gradient-driven optimization, supporting high-accuracy and high-quality assessment of adversarial robustness in NLP. Specifically, we address the aforementioned challenges in a unified optimization framework. And we develop an effective convex relaxation method to co-optimize the continuously-relaxed site selection and perturbation variables, and leverage an effective sampling method to establish an accurate mapping from the continuous optimization variables to the discrete textual perturbations. Moreover, as a first-order attack generation method, TextGrad can be baked in adversarial training to further improve the robustness of NLP models. Extensive experiments are provided to demonstrate the effectiveness of TextGrad not only in attack generation for robustness evaluation but also in adversarial defense. From the attack perspective, we show that TextGrad achieves remarkable improvements in both the attack success rate and the perplexity score over five state-of-the-art baselines. From the defense perspective, TextGrad-enabled adversarial training yields the most robust NLP model against a wide spectrum of NLP attacks.

## 1 Introduction

The assessment of adversarial robustness of machine learning (ML) models has received increasing research attention because of their vulnerability to adversarial input perturbations (known as adversarial attacks) (Goodfellow et al., 2014; Carlini & Wagner, 2017; Papernot et al., 2016). Among a variety of robustness evaluation methods, gradient-based adversarial attack generation makes a tremendous success in the computer vision (CV) domain (Croce & Hein, 2020; Dong et al., 2020). For example, the projected gradient descent (PGD)-based methods have been widely used to benchmark the adversarial robustness of CV models (Madry et al., 2018; Zhang et al., 2019b; Shafahi et al., 2019; Wong et al., 2020; Zhang et al., 2019a; Athalye et al., 2018). However, in the natural language processing (NLP) area, the predominant robustness evaluation tool belongs to query-based attack generation methods (Li et al., 2020; Jin et al., 2020; Ren et al., 2019; Garg & Ramakrishnan, 2020; Li et al., 2019), which do *not* make the full use of gradient information.

Yet, the (query-based) mainstream of NLP robustness evaluation suffers several limitations. First, these query-based attack methods could be prone to generating ambiguous or invalid adversarial textual inputs (Wang et al., 2021), most of which change the original semantics and could even

---

*Contributed equally.

Table 1: Effectiveness of TEXTGRAD at-a-glance on the SST-2 dataset (Socher et al., 2013) against 5 NLP attack baselines. Each attack method is categorized by the attack principle (*gradient*-based vs. *query*-based), and is evaluated at three aspects: *attack success rate (ASR)*, adversarial texts quality (in terms of language model *perplexity*), and runtime efficiency (averaged runtime for attack generation in seconds). Two types of victim models are considered, *i.e.*, realizations of BERT achieved by standard training (*ST*) and adversarial training (*AT*), respectively. Here AT integrates TEXTFOOLER (Jin et al., 2020) with standard training. Across models, higher ASR, lower perplexity, and lower runtime indicate stronger attack. The best performance is highlighted in **bold** per metric.

| Attack | Venue | Principle | | Attack Success Rate | | Perplexity | | Runtime Efficiency (s) |
| --- | --- | --- | --- | --- | --- | --- | --- | --- |
| | | Gradient | Query | ST | AT | ST | AT | |
| Jin et al. (2020) | AAAI | | ● | 82.8% | 43.6% | 431.4 | 495.7 | **1.03** |
| Li et al. (2020) | EMNLP | | ● | 86.4% | 76.4% | 410.7 | 357.8 | 1.69 |
| Garg & Ramakrishnan (2020) | EMNLP | | ● | 86.6% | 77.5% | 286.6 | 302.9 | 1.15 |
| Guo et al. (2021) | EMNLP | ● | | 85.7% | 79.7% | 314.0 | 381.7 | 11.44 |
| Lee et al. (2022) | ICML | | ● | 86.0% | 65.8% | 421.0 | 554.9 | 9.41 |
| TEXTGRAD (ours) | - | ● | | **93.5%** | **81.8%** | **266.4** | **285.3** | 3.65 |

mislead human annotators. Second, the query-based methods could be hardly integrated with the first-order optimization-based model training recipe, and thus makes it difficult to develop adversarial training-based defenses (Madry et al., 2018; Athalye et al., 2018). Even though some first-order optimization-based NLP attack generation methods were developed in the literature, they often come with poor attack effectiveness (Ebrahimi et al., 2018) or high computational cost (Guo et al., 2021), leaving the question of whether the best optimization framework for NLP attack generation is found. The most relevant work to ours is GBDA attack (Guo et al., 2021), which perturbs each token in the input by sampling substitutes from the whole vocabulary of victim model. The sample distribution is optimized using gradients to generate adversarial examples, but yields low computational efficiency and high memory cost. Inspired by above, we ask: *How to develop a principled gradient-based attack framework in NLP, like PGD in CV?*

The main challenges for leveraging gradients to generate adversarial attacks in NLP lie in two aspects. *First*, the discrete nature of texts makes it difficult to directly employ the gradient information on the inputs. Different from perturbing pixels in imagery data, adversarial perturbations in an textual input need to optimize over the discrete space of words and tokens. *Second*, the fluency requirement of texts imposes another constraint for optimization. In contrast to $\ell_p$-norm constrained attacks in CV, adversarial examples in NLP are required to keep a low perplexity score. The above two obstacles make the design of gradient-based attack generation method in NLP highly non-trivial.

To bridge the adversarial learning gap between CV and NLP, we develop a novel adversarial attack method, termed **TEXTGRAD**, by peering into gradient-driven optimization principles needed for effective attack generation in NLP. Specifically, we propose a convex relaxation method to co-optimize the perturbation position selection and token modification. To overcome the discrete optimization difficulty, we propose an effective sampling strategy to enable an accurate mapping from the continuous optimization space to the discrete textual perturbations. We further leverage a perplexity-driven loss to optimize the fluency of the generated adversarial examples. In **Table 1**, we highlight the attack improvement brought by TEXTGRAD over some widely-used NLP attack baselines. More thorough experiment results will be provided in Sec. 5.

**Our contribution.** ❶ We propose TEXTGRAD, a novel first-order gradient-driven adversarial attack method, which takes a firm step to fill the vacancy of a principled PGD-based robustness evaluation framework in NLP. ❷ We identify a few missing optimization principles to boost the power of gradient-based NLP attacks, such as convex relaxation, sampling-based continuous-to-discrete mapping, and site-token co-optimization. ❸ We also show that TEXTGRAD is easily integrated with adversarial training and enables effective defenses against adversarial attacks in NLP. ❹ Lastly, we conduct thorough experiments to demonstrate the superiority of TEXTGRAD to existing baselines in both adversarial attack generation and adversarial defense.

## 2 BACKGROUND AND RELATED WORK

**Adversarial attacks in CV.** Gradient information has played an important role in generating adversarial examples, *i.e.*, human-imperceptible perturbed inputs that can mislead models, in the CV

area (Goodfellow et al., 2014; Carlini & Wagner, 2017; Croce & Hein, 2020; Madry et al., 2018; Zhang et al., 2019b; Kurakin et al., 2016; Xu et al., 2019a;c). PGD attack is one of the most popular adversarial attack methods (Madry et al., 2018; Croce & Hein, 2020), which makes use of the first-order gradient to generate perturbations on the inputs and has achieved great success in attacking CV models with a low computational cost. Besides, PGD is also a powerful method to generate transfer attacks against unseen victim models (Szegedy et al., 2013; Liu et al., 2016; Moosavi-Dezfooli et al., 2017). Even in the black-box scenario (*i.e.*, without having access to model parameters), PGD is a principled framework to generate black-box attacks by leveraging function query-based gradient estimates (Cheng et al., 2018) or gradients of surrogate models (Cheng et al., 2019; Dong et al., 2018; Xie et al., 2019; Zou et al., 2020; Dong et al., 2019).

**Adversarial attacks in NLP.** Different from attacks against CV models, gradient-based attack generation methods are less popular in the NLP domain. HOTFLIP (Ebrahimi et al., 2018) is one of the most representative gradient-based attack methods by leveraging gradients to estimate the impact of character and/or word-level substitutions on NLP models. However, HOTFLIP neglects the optimality of site selection in the discrete character/token space and ignores the constraint on the post-attacking text fluency (to preserve readability) (Ren et al., 2019). By contrast, this work co-optimizes the selections of perturbation sites and tokens, and leverages a perplexity-guided loss to maintain the fluency of adversarial texts. Another attack method GBDA (Guo et al., 2021) models the token replacement operation as a probability distribution which is optimized using gradients. However, acquiring this probability distribution is accompanied with high computation and memory costs. By contrast, our work can achieve comparable or better performance with higher efficiency.

The mainstream of robustness evaluation in NLP in fact belongs to query-based methods (Li et al., 2020; Jin et al., 2020; Ren et al., 2019; Garg & Ramakrishnan, 2020; Li et al., 2019; Wang et al., 2021; Li et al., 2021b). Many current word-level query-based attack methods adopt a **two-phase** framework (Zang et al., 2020) including **(1)** generating candidate substitutions for each word in the input sentence and **(2)** replacing original words with the found candidates for attack generation.

*Phase (1)* aims at generating semantically-preserving candidate substitutions for each word in the original sentence. Genetic Attack (GA) (Alzantot et al., 2018) uses the word embedding distance to select the candidate substitutes and filter out the candidates with an extra language model. Such strategy is also adopted in many other attack methods like Jin et al. (2020) and Ebrahimi et al. (2018). PWWS (Ren et al., 2019) adopts WordNet (Miller, 1995) for candidate substitution generation. Similarly, PSO Attack (Zang et al., 2020) employs a knowledge base known as HowNet (Qi et al., 2019) to craft the candidate substitutions. Along with the development of the large pre-trained language models, Garg & Ramakrishnan (2020) and Li et al. (2020) propose to utilize mask language models such as BERT (Devlin et al., 2019) to predict the candidate substitutions. In TEXTGRAD, we also adopt the pre-trained language models to generate candidate substitutions.

*Phase (2)* requires the adversary to find a substitution combination from the candidates obtained in phase (1) to fool the victim model. A widely-used searching strategy is greedy search (Li et al., 2020; Jin et al., 2020; Ren et al., 2019; Garg & Ramakrishnan, 2020; Li et al., 2019), where each candidate is first ranked based on its impact on model prediction, and then the top candidate for each word is selected as the substitution. Another popular searching strategy is leveraging population-based methods, such as genetic algorithms and particle swarm optimization algorithms (Kennedy & Eberhart, 1995), to determine substitutions (Zang et al., 2020; Alzantot et al., 2018). Despite effectiveness, these query-based attack are prone to generating invalid or ambiguous adversarial examples (Wang et al., 2021) which change the original semantics and could even mislead humans. To overcome these problems, we propose TEXTGRAD, which leverages gradient information to co-optimize the perturbation position and token selection subject to a sentence-fluency constraint. We will empirically show that TEXTGRAD yields a better attack success rate with lower sentence perplexity compared to the state-of-the-art query-based attack methods.

**Adversarial training.** Adversarial training (AT) (Goodfellow et al., 2014; Madry et al., 2018) has been shown as an effective solution to improving robustness of deep models against adversarial attacks (Athalye et al., 2018). AT, built upon min-max optimization, has also inspired a large number of robust training approaches, ranging from supervised learning (Wong et al., 2020; Zhang et al., submitted to NeurIPS, 2021) to semi-supervised learning (Zhang et al., 2019b; Carmon et al., 2019), and further to self-supervised learning (Chen et al., 2020). Yet, the aforementioned literature focuses on AT for vision applications. Despite some explorations, AT for NLP models is generally under-

explored. In (Jin et al., 2020; Zang et al., 2020; Alzantot et al., 2018; Zhang et al., 2019c; Meng & Wattenhofer, 2020; Li et al., 2021a), adversarial data are generated offline and then integrated with the vanilla model training. In Li et al. (2021b); Zhu et al. (2019); Dong et al. (2021); Wang et al. (2020), the min-max based AT is adopted, but the adversarial attack generation step (*i.e.*, the inner maximization step) is conducted on the embedding space rather than the input space. As a result, both methods are not effective to defend against strong NLP attacks like TEXTGRAD.

## 3 MATHEMATICAL FORMULATION OF NLP ATTACKS

In this section, we start with a formal setup of NLP attack generation by considering two optimization tasks simultaneously: *(a)* (token-wise) attack site selection, and *(b)* textual perturbation via token replacement. Based on this setup, we will then propose a generic discrete optimization framework that allows for first-order gradient-based optimization.

**Problem setup.** Throughout the paper, we will focus on the task of text classification, where $\mathcal{M}(\mathbf{x})$ is a victim model targeted by an adversary to perturb its input $\mathbf{x}$. Let $\mathbf{x} = [x_1, x_2, \ldots, x_L] \in \mathbb{N}^L$ be the input sequence, where $x_i \in \{0, 1, \ldots, |V| - 1\}$ is the index of $i$th token, $V$ is the vocabulary table, and $|V|$ refers to the size of the vocabulary table.

From the *prior knowledge* perspective, we assume that an adversary has access to the victim model (*i.e.*, white-box attack), similar to many existing adversarial attack generation setups in both CV and NLP domains (Carlini & Wagner, 2017; Croce & Hein, 2020; Madry et al., 2018; Szegedy et al., 2013). Besides, the adversary has prior knowledge on a set of token candidates for substitution at each position, denoted by $\mathbf{s}_i = \{s_{i1}, s_{i2}, \ldots, s_{im}\}$ at site $i$, where $s_{ij} \in \{0, 1, \ldots, |V| - 1\}$ denotes the index of the $j$th candidate token that the adversary can be used to replace the $i$th token in $\mathbf{x}$. Here $m$ is the maximum number of candidate tokens.

From the *attack manipulation* perspective, the adversary has two goals: determining the optimal attack site as well as seeking out the optimal substitute for the original token. Given this, we introduce *site selection variables* $\mathbf{z} = [z_1, \ldots, z_L]$ to encode the optimized attack site, where $z_i \in \{0, 1\}$ becomes 1 if the token site $i$ is selected and 0 otherwise. In this regard, an *attack budget* is given by the number of modified token sites, $\mathbf{1}^T \mathbf{z} \le k$, where $k$ is the upper bound of the budget. We next introduce token-wise *replacement variables* $\mathbf{u}_i = [u_{i1}, \ldots, u_{im}]$, associated with candidates in $\mathbf{s}_i$, where $u_{ij} \in \{0, 1\}$, and $\mathbf{1}^T \mathbf{u}_i = 1$ if $z_i = 1$. Then, the $i$th input token $x_i$ will be replaced by the candidate expressed by $\hat{s}_i(\mathbf{u}_i; \mathbf{s}_i) = \sum_j (u_{ij} \cdot s_{ij})$. Please note that there is only one candidate token will be selected (constrained through $\mathbf{1}^T \mathbf{u}_i = 1$). For ease of presentation, we will use $\hat{\mathbf{s}}$ to denote the *replacement-enabled perturbed input* with the same length of $\mathbf{x}$.

In a nutshell, an NLP attack can be described as the perturbed input example together with site selection and replacement constraints (cstr):

$$
\begin{aligned}
&\text{Perturbed input: } \mathbf{x}^{\text{adv}}(\mathbf{z}, \mathbf{u}; \mathbf{x}, \mathbf{s}) = (1 - \mathbf{z}) \circ \mathbf{x} + \mathbf{z} \circ \hat{\mathbf{s}} \\
&\text{Discrete variables: } \quad \mathbf{z} \in \{0, 1\}^L, \mathbf{u}_i \in \{0, 1\}^m, \forall i \\
&\text{Site selection cstr: } \mathbf{1}^T \mathbf{z} \le k, \qquad \text{Replacement cstr: } \mathbf{1}^T \mathbf{u}_i = 1, \forall i,
\end{aligned}
\tag{1}
$$

where $\circ$ denotes the element-wise multiplication. For ease of notation, $\mathbf{u}$ and $\mathbf{s}$ are introduced to denote the sets of $\{\mathbf{u}_i\}$ and $\{\mathbf{s}_i\}$ respectively, and the adversarial example $\mathbf{x}^{\text{adv}}$ is a function of site selection and token replacement variables ($\mathbf{z}$ and $\mathbf{u}$) as well as the prior knowledge on the input example $\mathbf{x}$ and the inventory of candidate tokens $\mathbf{s}$. Based on (1), we will next formulate the optimization problem for generating NLP attacks.

**Discrete optimization problem with convex constraints.** The main difficulty of formulating the NLP attack problem suited for efficient optimization is the presence of discrete (non-convex) constraints. To circumvent this difficulty, a common way is to *relax* discrete constraints into their convex counterparts (Boyd et al., 2004). However, this leads to an inexact problem formulation. To close this gap, we propose the following problem formulation with continuous convex constraints and an attack loss built upon the discrete projection operation:

$$
\begin{aligned}
&\underset{\tilde{\mathbf{z}}, \tilde{\mathbf{u}}}{\text{minimize}} \quad \ell_{\text{atk}}(\mathbf{x}^{\text{adv}}(\mathcal{B}(\tilde{\mathbf{z}}), \mathcal{B}(\tilde{\mathbf{u}}); \mathbf{x}, \mathbf{s})) \\
&\text{subject to} \quad \mathcal{C}_1: \tilde{\mathbf{z}} \in [\mathbf{0}, \mathbf{1}], \mathbf{1}^T \tilde{\mathbf{z}} \le k, \ \mathcal{C}_2: \tilde{\mathbf{u}} \in [\mathbf{0}, \mathbf{1}], \mathbf{1}^T \tilde{\mathbf{u}}_i = 1, \forall i,
\end{aligned}
\tag{2}
$$

where most notations are consistent with (1), $\mathcal{B}$ is the projection operation that projects the continuous variables onto the Boolean set, *i.e.*, $\mathbf{z} \in \{0,1\}^L$ and $\mathbf{u}_i \in \{0,1\}^m$ in (1), and we use $\tilde{\mathbf{z}}$ and $\tilde{\mathbf{u}}_i$ as the continuous relaxation of $\mathbf{z}$ and $\mathbf{u}_i$. As will be evident later, an efficient projection operation can be achieved by randomized sampling. The graceful feature of problem (2) is that the constraint sets $\mathcal{C}_1$ and $\mathcal{C}_2$ are convex, given by the intersection of a continuous box and affine inequality/equalities.

# 4 OPTIMIZATION FRAMEWORK OF NLP ATTACKS

In this section, we present the details of gradient-driven first-order optimization that can be successfully applied to generating NLP attacks. Similar to the attack benchmark–projected gradient descent (PGD) attack Madry et al. (2018)–used for generating adversarial perturbations of imagery data, we propose the PGD attack framework for NLP models. We will illustrate the design of PGD-based NLP attack framework from four dimensions: 1) acquisition of prior knowledge on inventory of candidate tokens, 2) attack loss type, 3) regularization scheme to minimize the perplexity of perturbed texts, and 4) input gradient computation.

**Prior knowledge acquisition: candidate tokens for substitution.** We first tackle how to generate candidate tokens used for input perturbation. Inspired by BERT-ATTACK (Li et al., 2020) and BAE-R (Garg & Ramakrishnan, 2020), we employ pre-trained masked language models (Devlin et al., 2019; Liu et al., 2019; Lan et al., 2020), denoted as $\mathcal{G}$, to generate the candidate substitutions. Specifically, given the input sequence $\mathbf{x}$, we first feed it into $\mathcal{G}$ to get the token prediction probability at each position without masking the input. Then we take the top-$m$ tokens at each position as the candidates. Please note that getting the token predictions at each position does not require masking the input. Using the original sentence as the input can make the computation more efficient (only one forward pass to get predictions for all positions). With a similar approach, it has been shown in (Li et al., 2020) that the generated candidates are more semantically consistent with the original one, compared to the approach using actual "`[mask]`" tokens. Note that TEXTGRAD is compatible with any other candidate tokens generating method, making it general for practical usage.

**Determination of attack loss.** Most existing NLP attack generation methods (Li et al., 2020; Jin et al., 2020; Ren et al., 2019; Alzantot et al., 2018; Li et al., 2021a) use the negative cross-entropy (CE) loss as the attack objective. However, the CE loss hardly tells whether or not the attack succeeds. And it would hamper the optimization efficiency when the attack objective is regularized by another textual fluency objective (which will be introduced later). Our rationale is that intuitively a sentence with more aggressive textual perturbations typically yields a higher attack success rate but a larger deviation from its original format. Thus, it is more likely to be less fluent.

A desirable loss for designing NLP attacks should be able to indicate the attack status (failure vs. success) and can automatically adjust the optimization focus between the success of an attack and the promotion of perturbed texts fluency. Spurred by the above, we choose the C&W-type attack loss (Carlini & Wagner, 2017):

$$\ell_{\mathrm{atk}}(\mathbf{x}^{\mathrm{adv}}) = \max\{Z_{t_0}(\mathbf{x}^{\mathrm{adv}}) - \max_{i \neq t_0} Z_i(\mathbf{x}^{\mathrm{adv}}), 0\}, \tag{3}$$

where $\mathbf{x}^{\mathrm{adv}}$ was introduced in (2), $t_0$ denotes the predicted class of the victim model against the original input $\mathbf{x}$, and $Z_i$ denotes the prediction logit of class $i$. In (3), the difference $Z_{t_0}(\mathbf{x}^{\mathrm{adv}}) - \max_{i \neq t_0} Z_i(\mathbf{x}^{\mathrm{adv}})$ characterizes the confidence gap between the original prediction and the incorrect prediction induced by adversarial perturbations. The key advantages of using (3) for NLP attack generation are two-fold. First, the success of $\mathbf{x}^{\mathrm{adv}}$ (whose prediction is distinct from the original model prediction) is precisely reflected by its zero loss value, *i.e.*, $\ell_{\mathrm{atk}}(\mathbf{x}^{\mathrm{adv}}) = 0$. Second, the attack loss (3) has the self-assessment ability since it will be automatically de-activated only if the attack succeeds, *i.e.*, $Z_{t_0}(\mathbf{x}^{\mathrm{adv}}) \leq \max_{i \neq t_0} Z_i(\mathbf{x}^{\mathrm{adv}})$. Such an advantage facilitates us to strike a graceful balance between the attack success rate and the texts perplexity rate after perturbations.

**Text fluency regularization.** We next propose a differentiable texts fluency regularizer to be jointly optimized with the C&W attack loss,

$$\ell_{\mathrm{reg}} = \sum_i z_i \sum_j u_{ij}(\ell_{\mathrm{mlm}}(s_{ij}) - \ell_{\mathrm{mlm}}(x_i)) = \sum_i z_i \sum_j u_{ij}\ell_{\mathrm{mlm}}(s_{ij}) - \sum_i z_i\ell_{\mathrm{mlm}}(x_i), \tag{4}$$

where the last equality holds since $\sum_j \mathbf{u}_{ij} = 1$. $\ell_{\mathrm{mlm}}(\cdot)$ indicates the masked language modeling loss (Devlin et al., 2019) which is widely used for measuring the contribution of a word to the sentence fluency. For example, $\ell_{\mathrm{mlm}}(s_{ij})$ measures new sentence fluency after changing the $i$th position

as its $j$th candidate. Smaller $\ell_{\mathrm{mlm}}(s_{ij})$ indicates better sentence fluency after the replacement. We compute the masked language model loss $\ell_{\mathrm{mlm}}(x_i)$ for $i$th token and minimize the increment of masked language model loss after replacement.

**Input gradient calculation.** The availability of the gradient of the attack objective function is the precondition for establishing the PGD-based attack framework. However, the presence of the Boolean operation $\mathcal{B}(\cdot)$ in (2) prevents us from gradient calculation. To overcome this challenge, we prepend a randomized sampling step to the gradient computation. The rationale is that the continuously relaxed variables $\tilde{\mathbf{z}}$ and $\tilde{\mathbf{u}}$ in (2) can be viewed as (element-wise) site selection and token substitution probabilities. In this regard, given the continuous values $\tilde{\mathbf{z}} = \tilde{\mathbf{z}}_{t-1}$ and $\tilde{\mathbf{u}} = \tilde{\mathbf{u}}_{t-1}$ obtained at the last PGD iteration $(t-1)$, for the current iteration $t$ we can achieve $\mathcal{B}(\cdot)$ through the following Monte Carlo sampling step:

$$[\mathcal{B}^{(r)}(\tilde{\mathbf{z}}_{t-1})]_i = \left\{ \begin{array}{ll} 1 & \text{with probability } \tilde{z}_{t-1,i} \\ 0 & \text{with probability } 1 - \tilde{z}_{t-1,i} \end{array} \right. , \tag{5}$$

where $[\mathcal{B}^{(r)}(\tilde{\mathbf{z}}_{t-1})]_i$ denotes the $i$th element realization of $\mathcal{B}(\tilde{\mathbf{z}}_{t-1})$ at the $r$-th random trial. We use $R$ to denote the total number of sampling rounds. The above sampling strategy can be similarly defined to achieve $\mathcal{B}^{(r)}(\tilde{\mathbf{u}}_{t-1})$. It is worth noting that a large $R$ reduces the variance of the random realizations of $\mathcal{B}(\tilde{\mathbf{z}}_{t-1})$ and can further help reduce the gradient variance. Our empirical experiments show that $R = 20$ suffices to warrant satisfactory attack performance. Based on (5), the gradient of the attack objective function in (2) is given by

$$\mathbf{g}_{1,t} := \frac{1}{R} \sum_{r=1}^{R} \nabla_{\tilde{\mathbf{z}}} \ell_{\mathrm{atk}}(\mathbf{x}^{\mathrm{adv}}(\mathbf{z}^{(r)}, \mathbf{u}^{(r)}; \mathbf{x}, \mathbf{s})), \quad \mathbf{g}_{2,t} := \frac{1}{R} \sum_{r=1}^{R} \nabla_{\tilde{\mathbf{u}}} \ell_{\mathrm{atk}}(\mathbf{x}^{\mathrm{adv}}(\mathbf{z}^{(r)}, \mathbf{u}^{(r)}; \mathbf{x}, \mathbf{s})), \tag{6}$$

where $\mathbf{z}^{(r)} = \mathcal{B}^{(r)}(\tilde{\mathbf{z}}_{t-1})$ and $\mathbf{u}^{(r)} = \mathcal{B}^{(r)}(\tilde{\mathbf{u}}_{t-1})$, and $\mathbf{g}_{1,t}$ and $\mathbf{g}_{2,t}$ corresponds to the variables $\tilde{\mathbf{z}}$ and $\tilde{\mathbf{u}}$, respectively. Our gradient estimation over the discrete space also has the spirit similar to straight-through estimator (Bengio et al., 2013) and Gumbel Softmax method (Jang et al., 2016).

**Projected gradient descent (PGD) framework.** Based on the C&W attack loss (3), the texts fluency regularization (4) and the input gradient formula (6), we are then able to develop the PGD optimization method to solve problem (2). At the $t$-th iteration, PGD is given by

$$\tilde{\mathbf{z}}_t = \Pi_{\mathcal{C}_1}(\tilde{\mathbf{z}}_{t-1} - \eta_z \mathbf{g}_{1,t}), \quad \tilde{\mathbf{u}}_t = \Pi_{\mathcal{C}_2}(\tilde{\mathbf{u}}_{t-1} - \eta_u \mathbf{g}_{2,t}), \tag{7}$$

where $\Pi_{\mathcal{C}}$ denotes the Euclidean projection onto the constraint set $\mathcal{C}$, *i.e.*, $\Pi_{\mathcal{C}}(\mathbf{a}) = \arg\min_{\mathbf{x} \in \mathcal{C}} \|\mathbf{x} - \mathbf{a}\|_2^2$, and the constraint sets $\mathcal{C}_1$ and $\mathcal{C}_2$ have been defined in (2). Due to the special structures of these constraints, the closed forms of the projection operations $\Pi_{\mathcal{C}_1}$ and $\Pi_{\mathcal{C}_2}$ are attainable and illustrated in Proposition A.1 in the appendix. Using the optimization framework above, the empirical convergence of the PGD is relatively fast. It will be shown later, 5-step PGD is sufficient to make our algorithm outperforming all other baseline methods.

## 5 EXPERIMENTS

### 5.1 EXPERIMENT SETUP

**Datasets and attack baselines.** We mainly consider the following tasks[1]: SST-2 (Socher et al., 2013) for sentiment analysis, MNLI (Williams et al., 2018), RTE (Wang et al., 2018), and QNLI (Wang et al., 2018) for natural language inference and AG News (Zhang et al., 2015) for text classification. We compare our proposed TEXTGRAD method with the following state-of-the-art white box and black-box NLP **attack baselines**: TEXTFOOLER(Jin et al., 2020), BERT-ATTACK (Li et al., 2020), BAE-R (Garg & Ramakrishnan, 2020), HOTFLIP (Ebrahimi et al., 2018), BBA (Lee et al., 2022) and GBDA (Guo et al., 2021). We also include a greedy search-based method termed GREEDY, which combines the candidate genration method used in ours and the Greedy-WIR search strategy in Morris et al. (2020). In this regard, since the candidate substitute set is the same for GREEDY and TEXTGRAD, we can better demonstrate the advantage of TEXTGRAD over baselines that use greedy search to craft adversarial examples. We follow the benchmark attack setting in (Wang et al., 2021; Li et al., 2021b). The attack budget is set to 25% of the total word numbers in a sentence for baselines and TEXTGRAD to ensure the fair comparison. More details about the attack implementations could be seen in Appendix B.

---

[1]Codes are available at `https://github.com/UCSB-NLP-Chang/TextGrad`

**Victim models.** We consider two classes of victim models in experiments, namely conventionally trained standard models and robustly trained models with awareness of adversarial attacks. In the robust training paradigm, we consider Adversarial Data Augmentation (ADA), Mixup-based Adversarial Data Augmentation (MIXADA) (Si et al., 2021), PGD-AT (Madry et al., 2018), FREELB (Zhu et al., 2019), INFOBERT (Wang et al., 2020), and ASCC (Dong et al., 2021). Notably, except ADA and MIXADA, other robust training methods impose adversarial perturbations at the (continuous) embedding space. Following (Li et al., 2021b), we remove the $\ell_2$ perturbation constraint when training with PGD-AT and FREELB.

All the victim models are fine-tuned based on three popular NLP encoders, *i.e.*, BERT-base (Devlin et al., 2019), RoBERTa-large (Liu et al., 2019), and ALBERT-xxlargev2 (Lan et al., 2020). When attacking these models, TEXTGRAD use the corresponding masked language model to generate candidate substitutes. We also follow the best training settings in (Li et al., 2021b).

**Evaluation metrics.** First, attack success rate (ASR) measures the attack effectiveness, given by the number of examples that are successfully attacked over the total number of attacked examples. Second, perplexity (PPL) measures the quality of the generated adversarial texts. We use the pre-trained GPT-XL (Radford et al., 2019) language model for PPL evaluation.

Table 2: Performance of proposed TEXTGRAD attack method and baseline methods against normally trained victim models. The performance is measured by attack success rate (ASR) as well as perplexity (PPL) across different datasets and model architectures. A more powerful attack method is expected to have a higher (↑) ASR and lower (↓) PPL. The **best** results under each metric are highlighted in **bold** and second best are underlined.

| Dataset | Attack Method | BERT ASR | BERT PPL | RoBERTa ASR | RoBERTa PPL | ALBERT ASR | ALBERT PPL |
|---|---|---|---|---|---|---|---|
| SST-2 | TEXTFOOLER | 82.84 | 431.44 | 69.38 | 483.29 | 69.77 | 536.34 |
| | BERT-ATTACK | 86.44 | 410.72 | 79.34 | 435.29 | 82.73 | 419.78 |
| | BAE-R | 86.62 | 286.63 | 85.92 | 300.08 | 85.80 | **293.29** |
| | GREEDY | 87.79 | 427.73 | 91.12 | 408.12 | 88.47 | 432.29 |
| | HOTFLIP | 56.07 | 277.79 | 23.30 | **196.81** | 18.35 | 293.53 |
| | BBA | 85.96 | 421.00 | 81.51 | 532.69 | 81.19 | 461.60 |
| | GBDA | 85.70 | 314.00 | - | - | - | - |
| | TEXTGRAD | **93.51** | **266.41** | **96.45** | 274.90 | **93.51** | 313.64 |
| MNLI-m | TEXTFOOLER | 74.82 | 320.47 | 67.33 | 314.11 | 66.02 | 322.00 |
| | BERT-ATTACK | 88.77 | 234.53 | 86.78 | 241.25 | 85.16 | 246.35 |
| | BAE-R | 87.00 | 196.20 | 84.56 | 191.29 | 85.39 | **223.62** |
| | GREEDY | 88.17 | 263.47 | 90.43 | 265.53 | 88.66 | 272.94 |
| | HOTFLIP | 54.44 | 276.32 | 26.36 | 204.72 | 29.14 | 316.93 |
| | BBA | 82.86 | 346.60 | 78.44 | 329.63 | 77.84 | 423.60 |
| | GBDA | 93.37 | 290.41 | - | - | - | - |
| | TEXTGRAD | **94.08** | **193.42** | **95.44** | **211.58** | **94.44** | 264.07 |
| RTE | TEXTFOOLER | 59.55 | 402.44 | 62.50 | 319.64 | 74.31 | 344.40 |
| | BERT-ATTACK | 64.61 | 329.30 | 74.57 | 279.86 | 79.17 | 343.69 |
| | BAE-R | 65.73 | 239.68 | 71.21 | 221.44 | 81.94 | **317.96** |
| | GREEDY | 60.67 | 501.40 | 78.81 | 228.03 | 82.52 | 517.97 |
| | HOTFLIP | 45.51 | 318.13 | 70.25 | 184.47 | 34.97 | 801.32 |
| | BBA | 60.67 | 361.30 | 69.16 | 239.41 | 70.62 | 418.07 |
| | GBDA | 68.20 | 471.20 | - | - | - | - |
| | TEXTGRAD | **71.91** | **202.96** | **83.90** | **140.51** | **87.41** | 378.07 |
| QNLI | TEXTFOOLER | 53.55 | 399.90 | 48.17 | 398.45 | 58.34 | 451.02 |
| | BERT-ATTACK | 63.86 | 384.28 | 60.11 | 376.25 | 64.31 | 411.93 |
| | BAE-R | 62.31 | 324.14 | 60.86 | 309.45 | 62.81 | 324.14 |
| | GREEDY | 67.95 | 443.61 | 63.74 | 462.42 | 62.71 | 379.64 |
| | HOTFLIP | 48.07 | 301.35 | 49.91 | 313.47 | 44.27 | 383.42 |
| | BBA | 60.31 | 498.74 | 59.12 | 429.77 | 58.74 | 461.55 |
| | GBDA | 63.52 | 1473.15 | - | - | - | - |
| | TEXTGRAD | **70.48** | **297.59** | **68.00** | **297.16** | **72.43** | 333.24 |
| AG News | TEXTFOOLER | 59.43 | 486.53 | 60.77 | 427.19 | 64.37 | 475.62 |
| | BERT-ATTACK | 62.41 | 560.90 | 63.08 | 513.24 | 68.42 | 496.92 |
| | BAE-R | 67.97 | 519.42 | 68.79 | 527.68 | 72.73 | 374.35 |
| | GREEDY | 60.35 | 523.64 | 62.35 | 579.84 | 73.25 | 375.42 |
| | HOTFLIP | 49.27 | 397.60 | 52.08 | 375.46 | 55.41 | 431.36 |
| | BBA | **74.70** | **147.22** | 66.03 | **157.49** | 55.61 | 323.71 |
| | GBDA | 70.28 | 456.60 | - | - | - | - |
| | TEXTGRAD | 74.51 | 303.21 | **75.19** | 303.91 | **85.12** | 397.93 |

[1] The official implementation of GBDA does not support attacking RoBERTa/ALBERT. See Appendix B for detailed explanations.

## 5.2 EXPERIMENT RESULTS

**Attack performance on normally-trained standard models.** In Table 2, we compare the attack performance (in terms of ASR and PPL) of TEXTGRAD with **7** NLP attack baselines across **5** datasets and **3** victim models that are normally trained. As we can see, TEXTGRAD yields a better ASR than all the other baselines, leading to at least 3% improvement in nearly all settings. Except our method, there does not exist any baseline that can win either across dataset types or across model types. From the perspective of PPL, TEXTGRAD nearly outperforms all the baselines when attacking BERT and RoBERTa. For the victim model ALBERT, TEXTGRAD yields the second or the third best PPL, with a small performance gap to the best PPL result. Additionally, the ASR improvement gained by TEXTGRAD remains significant in all settings when attacking ALBERT, which shows a good adversarial robustness in several past robustness evaluations (Wang et al., 2021).

**Attack performance on robustly-trained models.** Table 3 demonstrates the attack effectiveness of TEXTGRAD against robustly-trained models. To make a thorough assessment, attack methods are compared under **6** robust BERT models obtained using **6** robust training methods on SST-2 and RTE datasets. As we can see, TEXTGRAD yields the best ASR in all settings. Among baselines, BBA and GBDA seem outperforming the others when attacking robust models. However, compared to TEXTGRAD, there remains over 4% ASR gap in most of cases. From the PPL perspective, TEXTGRAD achieves at least the second best performance. It is worth noting that the considered test-time attacks (including TEXTGRAD and baselines) are not seen during robust training. Therefore, all the robust models are not truly robust when facing unforeseen attacks. In particular, TEXTGRAD can easily break these defenses on SST-2, as evidenced by achieving at least 89% ASR.

Table 3: Performance of attack methods against robustly-trained victim models. Different robustified versions of BERT are obtained using different adversarial defenses including ADA, MixADA (Si et al., 2021), PGD-AT (Madry et al., 2018), Free-LB (Zhu et al., 2019), InfoBERT (Wang et al., 2020), and ASCC (Dong et al., 2021). Two datasets including SST-2 and RTE are considered. The attack performance is measured by ASR and PPL. The **best** results under each metric (corresponding to each column) are highlighted in **bold** and the second best results are underlined.

| Dataset | Attack Method | ADA | | MixADA | | PGD-AT | | Free-LB | | InfoBERT | | ASCC | |
|---|---|---|---|---|---|---|---|---|---|---|---|---|---|
| | | ASR | PPL | ASR | PPL | ASR | PPL | ASR | PPL | ASR | PPL | ASR | PPL |
| SST-2 | TextFooler | 81.20 | 550.91 | 81.56 | 531.24 | 74.30 | 481.40 | 71.99 | 474.86 | 80.19 | 440.09 | 83.46 | 554.42 |
| | Bert-Attack | 84.77 | 455.21 | 84.47 | 459.72 | 81.38 | 408.57 | 79.45 | 407.13 | 86.52 | 443.85 | 83.33 | 431.65 |
| | BAE-R | 85.90 | 324.66 | 86.90 | **291.64** | 83.22 | 288.92 | 79.86 | 341.96 | 86.22 | 324.54 | 83.46 | 296.65 |
| | Greedy | 86.20 | 447.78 | 85.30 | 433.93 | 80.79 | 411.48 | 78.57 | 440.61 | 87.40 | 429.34 | 80.52 | 406.83 |
| | HotFlip | 47.11 | 353.28 | 49.26 | 365.85 | 40.87 | 353.18 | 23.84 | 299.68 | 48.79 | 355.40 | 61.55 | 354.29 |
| | BBA | 90.00 | 510.83 | 88.37 | 479.33 | 78.94 | 464.86 | 83.61 | 463.33 | 84.86 | 479.11 | 91.69 | 455.72 |
| | GBDA | 83.40 | 321.75 | 86.12 | 346.70 | 83.87 | 368.14 | 67.35 | 325.05 | 83.50 | 274.11 | 27.71 | 318.89 |
| | TextGrad | **93.81** | 316.85 | **93.72** | 324.24 | **89.00** | **271.32** | **92.20** | **272.11** | **93.00** | **270.37** | **92.00** | **255.78** |
| RTE | TextFooler | 62.42 | 300.29 | 65.62 | 384.39 | 56.22 | 214.87 | 63.40 | 270.23 | 68.36 | 405.06 | 49.41 | 278.88 |
| | Bert-Attack | 68.48 | 296.96 | 71.88 | 264.10 | 60.54 | 287.78 | 64.95 | 313.94 | 72.88 | 365.47 | 55.88 | 313.11 |
| | BAE-R | 66.67 | 271.80 | 72.50 | **230.12** | 61.62 | **199.79** | 67.53 | 274.13 | 69.49 | 293.98 | 52.35 | 216.94 |
| | Greedy | 63.86 | 378.57 | 73.46 | 432.57 | 58.60 | 392.67 | 68.88 | 463.37 | 71.35 | 372.42 | 55.03 | 435.63 |
| | HotFlip | 30.12 | 479.65 | 25.31 | 379.64 | 29.57 | 405.00 | 46.94 | 368.25 | 28.65 | 345.80 | 33.73 | 290.61 |
| | BBA | 69.69 | 340.60 | 70.65 | 418.07 | 62.16 | 271.51 | 66.49 | 419.29 | 66.10 | 358.07 | 52.35 | 237.69 |
| | GBDA | **81.81** | 986.36 | 75.30 | 473.80 | 67.02 | 973.50 | 70.10 | 1461.07 | **84.18** | 1331.04 | 34.70 | 483.38 |
| | TextGrad | 80.12 | **219.81** | **87.04** | 254.19 | **71.51** | 223.56 | **78.06** | **302.81** | 82.58 | **204.64** | **67.46** | 226.54 |

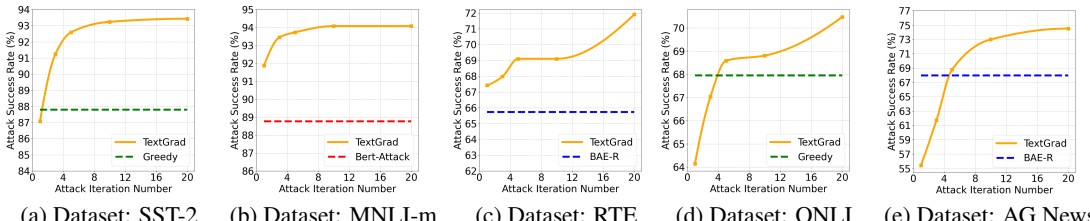

(a) Dataset: SST-2    (b) Dataset: MNLI-m    (c) Dataset: RTE    (d) Dataset: QNLI    (e) Dataset: AG News

Figure 1: ASR of TextGrad with different attack iteration numbers. We attack the standard BERT model with TextGrad on different datasets. For each dataset, we show the curve of ASR of TextGrad *w.r.t* different iteration numbers (the orange curves) as well as the ASR of the best query-based baseline on each corresponding dataset from Table 2 (the dashed lines). TextGrad can beat the best baseline with only 5 iterations on all datasets.

**TextGrad versus attack strengths.** Moreover, we evaluate the attack performance of TextGrad from two different attack strength setups: *(a)* the number of optimization steps for attack generation, and *(b)* the maximum number of words modifications (that we call attack budget), *i.e.*, $k$ in (2). Results associated with above setups (a) and (b) are presented in Figure 1 and Table 4. In both cases, we compare the performance of TextGrad with that of baselines when attacking a normally trained BERT model. As shown in Figure 1, ASR of TextGrad increases as the number of attack

Table 4: Effect of attack budget $k$ on ASR of TextGrad. Evaluation is performed under the standard BERT model on SST. Recall that the attack budget constrains the ratio of the words allowed to be modified in a sentence. Here $k = x\%$ is short for $k = x\%$ of the number of words in a sentence.

| Attack Method | k = 5% | k = 10% | k = 15% | k = 20% | k = 25% | k = 30% |
|---|---|---|---|---|---|---|
| TextFooler | 14.56 | 42.69 | 62.79 | 75.41 | 82.84 | 86.67 |
| Bert-Attack | 19.40 | 43.99 | 72.35 | 84.85 | 86.44 | 92.03 |
| BAE-R | 20.11 | 53.60 | 70.46 | 81.31 | 86.62 | 89.03 |
| Greedy | 18.04 | 49.23 | 68.46 | 80.84 | 87.79 | 90.07 |
| HotFlip | 9.26 | 27.48 | 39.80 | 49.65 | 56.07 | 59.96 |
| BBA | 17.45 | 48.29 | 68.16 | 79.36 | 85.96 | 89.59 |
| TextGrad | **31.56** | **56.78** | **77.73** | **88.74** | **93.51** | **95.81** |

iterations increases. Moreover, TextGrad achieves a substantial ASR improvement over the best baseline by merely taking a very small number of iterations (less than 8 iterations in all cases). As shown in Table 4, ASR of TextGrad increases as the attack budget ($k$) increases. Additionally, even if $k$ is small (*e.g.*, $k = 5\%$ of the number of words), TextGrad still significantly outperforms the baselines in ASR.

**Other experiment results.** In Appendix C-E, we further include the *attack transferability*, *ablation study*, and *human evaluation*. In a nutshell, we show that TextGrad achieves graceful attack transferability. The optimization techniques including random restarting and randomized sampling help boost the attack performance. For human evaluation, TextGrad outperforms BAE-R, which performs the best in automatic quality evaluation.

Table 5: Robustness evaluation of different adversarial training methods on SST-2 dataset. The performance is measured by clean accuracy (%) and robust accuracy (%) under different attack types. We also report the average robust accuracy (Avg RA) over different attack types.

| Model | Clean Accuracy | Attack Method | | | | | Avg RA |
| --- | --- | --- | --- | --- | --- | --- | --- |
| | | TEXTGRAD | TEXTFOOLER | BERT-ATTACK | BAE-R | AdvGLUE | |
| ST | 93.14 | 6.04 | 15.98 | 12.63 | 12.47 | 30.55 | 15.53 |
| PGD-AT | 92.31 | 10.15 | 23.72 | 12.52 | 15.49 | 36.80 | 19.73 |
| FREE-LB | 93.52 | 7.30 | 26.19 | 14.66 | 18.84 | 27.77 | 18.96 |
| INFOBERT | 92.86 | 6.47 | 18.40 | 9.28 | 12.80 | 28.47 | 15.08 |
| ASCC | 87.94 | 7.02 | 14.55 | 14.66 | 14.55 | 40.27 | 18.21 |
| TEXTFOOLER-AT | 88.16 | 16.03 | 49.70 | 20.81 | 19.82 | 54.86 | 32.24 |
| TEXTGRAD-AT | 80.40 | 50.58 | 33.94 | 27.62 | 41.13 | 53.47 | 41.34 |
| TEXTGRAD-TRADES | 81.49 | 55.91 | 35.53 | 32.02 | 39.43 | 51.38 | 42.85 |

## 6 ADVERSARIAL TRAINING WITH TEXTGRAD

In this section, we empirically show that TEXTGRAD-based AT (termed TEXTGRAD-AT) provides an effective adversarial defense comparing to other AT baselines in NLP. We focus on robustifying a BERT model on SST-2 dataset, and set the train-time iteration number of TEXTGRAD to 5 and restart time to 1. During evaluation, we set TEXTGRAD with 20 attack iterations and 10 restarts.

Table 5 makes a detailed comparison between TEXTGRAD-AT with 6 baselines including ① standard training (ST), ② PGD-AT (Madry et al., 2018), ③ FREE-LB (Zhu et al., 2019), ④ INFOBERT (Wang et al., 2020), ⑤ ASCC (Dong et al., 2021), and ⑥ TEXTFOOLER-AT. We remark that the AT variants ②–⑤ generate adversarial perturbations against the continuous feature representations of input texts rather than the raw inputs at the training phase. By contrast, ours and TEXTFOOLER-AT generate adversarial examples in the discrete input space. As will be evident later, TEXTFOOLER-AT is also the most competitive baseline to ours. Besides TEXTGRAD-AT, we also propose TEXTGRAD-TRADES by integrating TEXTGRAD with TRADES (Zhang et al., 2019b), which is commonly used to optimize the tradeoff between accuracy and robustness. See more implementation details in Appendix B. At testing time, four types of NLP attacks (TEXTGRAD, TEXTFOOLER, BERT-ATTACK, and BAE-R) are used to evaluate the robust accuracy of models acquired by the aforementioned training methods.

Our key observations from Table 5 are summarized below. **First**, the models trained by TEXTGRAD-AT and TEXTGRAD-TRADES achieve the best robust accuracy in nearly all settings. The only exception is the case of TEXTFOOLER-AT vs. the TEXTFOOLER attack, since TEXTFOOLER is an unseen attack for our approaches but it is known for TEXTFOOLER-AT during training. By contrast, if non-TEXTGRAD and non-TEXTFOOLER attacks are used for robustness evaluation, then our methods outperform TEXTFOOLER-AT by a substantial margin (*e.g.*, robust enhancement under BAE-R). **Second**, we evaluate the performance of different robust training methods under the AdvGLUE (Wang et al., 2021) benchmark. We observe that TEXTGRAD-AT and TEXTGRAD-TRADES perform better than the baselines ②-⑤, while TEXTFOOLER-AT is slightly better than ours. However, this is predictable since AdvGlue uses TEXTFOOLER as one of the attack methods to generate the adversarial dataset (Wang et al., 2021). As shown by the average robust accuracy in Table 5, our methods are the only ones to defend against a wide spectrum of attack types. **Third**, our methods trade off an accuracy drop for robustness improvement. However, the latter is much more significant than the former. For example, none of baselines can defend against TEXTGRAD, while our improvement in robust accuracy is over 35% compared to TEXTFOOLER-AT. We further show that robust accuracy converges quickly in two epoch training in Appendix F, demonstrating that TEXTGRAD-AT can be used to enhance the robustness of downstream tasks in an efficient manner.

## 7 CONCLUSION

In this paper, we propose TEXTGRAD, an effective attack method based on gradient-driven optimization in NLP. TEXTGRAD not only achieves remarkable improvements in robustness evaluation but also boosts the robustness of NLP models with adversarial training. In the future, we will consider how to generalize TEXTGRAD to other types of perturbations, such as word insertion or deletion, to further improve the attack performance.

## 8 ACKNOWLEDGMENT

The work of Bairu Hou, Jinghan Jia, Yihua Zhang, Sijia Liu, and Shiyu Chang was partially supported by National Science Foundation (NSF) Grant IIS-2207052. The computing resources used in this work were partially supported by the MIT-IBM Watson AI Lab.

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

# A    PROJECTION OPERATIONS IN PGD FRAMEWORK

The projection operations $\Pi_{\mathcal{C}_1}$ and $\Pi_{\mathcal{C}_2}$ in Equation 4 are demonstrated below:

**Proposition A.1.** *Given $\mathcal{C}_1 = \{\tilde{\mathbf{z}}|1^T\tilde{\mathbf{z}} \leq k, \tilde{\mathbf{z}} \in [0,1]^L\}$, the project operation for a data point $\tilde{\mathbf{z}}'$ with respect to $\mathcal{C}_1$ is*

$$\Pi_{\mathcal{C}_1}(\tilde{\mathbf{z}}') = \begin{cases} P_{[0,1]}[\tilde{\mathbf{z}}'] & \text{if } 1^T P_{[0,1]}[\tilde{\mathbf{z}}'] \leq k, \\ P_{[0,1]}[\tilde{\mathbf{z}}' - \boldsymbol{\mu}\mathbf{1}] & \text{if } \boldsymbol{\mu} > 0 \text{ and } 1^T P_{[0,1]}[\tilde{\mathbf{z}}' - \boldsymbol{\mu}\mathbf{1}] = k, \end{cases} \tag{8}$$

*and for $\mathcal{C}_2 = \{\tilde{\mathbf{u}}_i|1^T\tilde{\mathbf{u}}_i = 1, \tilde{\mathbf{u}}_i \in [0,1]^m\}$, the project operation for a data point $\tilde{\mathbf{u}}_i'$ with respect to $\mathcal{C}_2$ is:*

$$\Pi_{\mathcal{C}_2}[\tilde{\mathbf{u}}_i'] = P_{[0,1]}[\tilde{\mathbf{u}}_i' - \boldsymbol{v}\mathbf{1}] \tag{9}$$

*where $\boldsymbol{v}$ is the roof of $\mathbf{1}^T P_{[0,1]}[\tilde{\mathbf{u}}_i' - \boldsymbol{v}\mathbf{1}] = 1$, $P_{[0,1]}(\cdot)$ is an element-wise projection operation, $P_{[0,1]}(x) = x$ if $x \in [0,1]$, $0$ if $x < 0$, and $1$ if $x > 1$.*

A similar derivation of the projection has been shown in (Xu et al., 2019b).

# B    IMPLEMENTATION DETAILS

In this section, we include the implementation details of victim models, hyper-parameters for baselines, and training details.

**Training of victim models**    We run our experiments on the Tesla V100 GPU with 16GB memory. We fine-tune the pre-trained BERT-base-uncased model on each dataset with a batch size of 32, a learning rate of $2e$-5 for 5 epochs. For RoBERTa-large and ALBERT-xxlargev2, we use a batch size of 16 and learning rate of $1e$-5. For the robust models, we use the implementation of (Li et al., 2021b). Each model is trained for 10 epochs with a learning rate of $2e$-5 and batch size of 32. Specifically, for ADA and MIXADA, we perturb the whole training dataset for augmentation. For MIXADA, we use $\alpha = 2.0$ in the beta distribution and mix the hidden representations sampled from layer $\{7, 9, 12\}$. For PGD-AT, we use step size $\alpha = 3e$-2 and number of steps $m = 5$ for both SST and RTE dataset. For FREE-LB, we use step size $\alpha = 1e$-1 and number of steps $m = 2$ on SST-2 dataset and $\alpha = 3e$-2, $m = 3$ on RTE dataset. For INFOBERT, the step size $\alpha$ is $4e$-2 and number of steps is 3 for both two datasets. Finally, we use $\alpha = 10, \beta = 4$ and run for 5 steps to generate perturbation for ASCC. For datasets that the labels of testing dataset are not available (MNLI, RTE, QNLI), we randomly sample 10% of training dataset as validation dataset and use the original validation dataset for testing. For the AG News dataset where the validation set is not available, we use the same way to generate the validation dataset.

**Attack Implementation**    Regarding the hyper-parameters of TEXTGRAD, we utilize 20-step PGD for optimization and fix the number of sampling $R$ in each iteration to be 20. We adopt a learning rate of 0.8 for both $\tilde{\mathbf{z}}$ and $\tilde{\mathbf{u}}$, and normalize the gradient $\mathbf{g}_{1,t}$ and $\mathbf{g}_{2,t}$ to unit norm before the descent step. After PGD optimization, we sample 20 different $\tilde{\mathbf{z}}$ and $\tilde{\mathbf{u}}$ pairs for a single input $\mathbf{x}$. To determine the final adversarial example of $\mathbf{x}$, we select the one with the minimum PPL measured by the GPT-2 language model (Radford et al., 2019) among all successful attacks. Although such a post-processing approach via multiple sampling introduces additional computational overhead, it ensures the high quality of generated attacks. If all 20 sampled attacks fail to fool the victim model, we allow TEXTGRAD to restart the attack process with a different initialization of $\tilde{\mathbf{z}}$ and $\tilde{\mathbf{u}}$. Restart with different initializations is a standard setting used in white-box PGD attacks for imagery data. Here, we set the maximum number of restarts to 10. However, empirically we find that most of the attacks will be successful with a single or without restart. The average number of restarts in our experiment is around 1.6-1.8. To ensure query-based baselines approaches with a large enough query budget, we set the maximum number of queries for them to be 2000.

**Attack parameters**    For baselines, the attack budget is set to 25% of the total word numbers in a sentence. Since TEXTGRAD modifies the sentence in token-level, we set the maximum tokens that TEXTGRAD can modify to be 25% of the total word numbers in a sentence. By doing

so, TEXTGRAD uses the same or less attack budget than baselines, and ensures the fair comparison. We use the implementation of the TextAttack (Morris et al., 2020) library for TEXTFOOLER, BERT-ATTACK, BAE-R and BBA. The number of candidate substitutions for each word is 50 in TEXTFOOLER and BAE-R. For BERT-ATTACK, we set this value to 48 following the default setting. For BBA we use the same candidate substitution generation method as TEXTFOOLER, which is consistent with the original paper. We use our candidate substitute generation method in HOTFLIP and GREEDY and the number of candidate substitution tokens for the two baselines is also 50. For natural language inference datasets (MNLI-m, QNLI, RTE), we restrict all the attackers to only perturb the hypothesis.

GBDA purely rely on soft constraints during attack instead of hard attack budgets (*i.e.*, the maximum perturbation which is 25% of the total word numbers in a sentence for all the other methods including TEXTGRAD). Furthermore, the soft constraints in GBDA rely on an extra causal language model (such as GPT-2 (Radford et al., 2019)) during attacking. When attacking masked language models such as BERT, RoBERTa, and ALBERT, one needs to train a causal language model that shares the same vocabulary with the victim model, making their method less flexible. Since the official implementation of GBDA only provides a causal language model that has the same vocabulary table as BERT, it only support attacking BERT and cannot be used to evaluation the robustness of RoBERTa/ALBERT in our experiments in Table 2. Therefore, we only report the experiment results when the victim model is BERT. We use the default hyper-parameters of GBDA in their original paper (100 attack iterations with batch size 10 and learning rate 0.3; $\lambda_{\text{perp}} = 1$). For $\lambda_{\text{sim}}$, we follow the paper's setting to cross-validate it in range $[20, 200]$. Finally, for SST-2 and MNLI-m datasets, we use $\lambda_{\text{sim}} = 50$; for other dataset we find the attack example quality is very low given a small $\lambda_{\text{sim}}$. Therefore we use $\lambda_{\text{sim}} = 200$ for the other datasets.

For TEXTGRAD, we also consider more attack constraints. Firstly, we conduct part-of-speech checking before attacking and only nouns, verbs, adjectives, and adverbs can be replaced. Secondly, after generating candidate substitutions, we use WordNet (Miller, 1995) to filter antonyms of original words to avoid invalid substitutions. Thirdly, stop-words will not be considered during attacking, which means we will not substitute original words that are stop-words or replace original words with stop-words. Finally, considering the tokenization operation in pre-trained language models, we filter out sub-words in the generated candidate substitutions before attacking to further improve the quality of adversarial examples.

## C  ATTACK TRANSFERABILITY

We compare the attack transferability of various attack methods in this section. Table 6 compares the attack transferability of various attack methods given different pairs of a source victim model used for attack generation and a targeted model used for transfer attack evaluation, where the considered models (BERT, RoBERTa, and ALBERT) are normally trained on SST-2. As we can see, transfer attacks commonly lead to the ASR degradation. However, compared to baselines, TEXTGRAD yields better transfer attack performance in nearly all the cases. It is also worth noting that there exists a large ASR drop when NLP attacks transfer to an unseen model. Thus, it requires more in-depth future studies to tackle the question of how to improve transferability of NLP attacks.

## D  ABLATION STUDIES

Table 7 demonstrates the usefulness of proposed optimization tricks: random restarting and randomized sampling, where the former has been commonly used in generating adversarial images (Madry et al., 2018), and the latter is critical to calculate input gradients. As we can see, both optimization tricks play positive roles in boosting the attack performance of TEXTGRAD. However, the use of randomized sampling seems more vital, as evidenced by the larger ASR drop (6%-14%) when using TEXTGRAD w/o randomized sampling.

We also study the performance of variants of baselines. Specifically, BBA can be combined with different substitution generation methods. In our main experiments, we evaluate the attack performance of BBA using the substitution generation method of TEXTFOOLER, which adopts the word embedding similarity to generate substitutions. In Table 8 we test the performance of BBA when using the substitution generation method of BAE-R.

Table 6: NLP attack transferability. Attacks generated from source victim models (BERT, RoBERTa, and ALBERT) are evaluated at the same set of models. The experiment is conducted on the SST-2 dataset and all the models are normally trained.

| Attack Method | Victim Model | Surrogate Model | | |
|---|---|---|---|---|
| | | BERT | RoBERTa | ALBERT |
| TEXTFOOLER | BERT | 82.84 | 37.83 | 37.84 |
| | RoBERTa | 25.47 | 69.38 | 31.07 |
| | ALBERT | 20.69 | 23.46 | 69.77 |
| BERT-ATTACK | BERT | 86.44 | 39.38 | 42.78 |
| | RoBERTa | 36.50 | 79.34 | 42.84 |
| | ALBERT | 32.02 | 33.14 | 82.73 |
| BAE-R | BERT | 86.62 | 55.86 | 57.10 |
| | RoBERTa | 55.54 | 85.92 | 57.55 |
| | ALBERT | 53.36 | 58.15 | 85.80 |
| Greedy | BERT | 87.79 | 59.70 | 57.48 |
| | RoBERTa | 47.52 | 90.43 | 57.77 |
| | ALBERT | 44.45 | 55.59 | 88.66 |
| TEXTGRAD | BERT | 93.51 | 64.92 | 55.13 |
| | RoBERTa | 54.10 | 96.45 | 57.39 |
| | ALBERT | 49.68 | 58.10 | 93.51 |

Table 7: Sensitivity analysis of random restarts and sampling. TEXTGRAD is conducted over a normally trained BERT model.

| Attack Method | SST-2 | MNLI-m | RTE | QNLI | AG News |
|---|---|---|---|---|---|
| TextGrad | **93.51** | **94.08** | **71.91** | **70.48** | **74.51** |
| *w.o.* restart | 90.27 | 91.90 | 67.42 | 65.26 | 68.87 |
| *w.o.* restart & sampling | 78.30 | 85.44 | 58.99 | 55.05 | 54.23 |

From the experiment results, we highlight the following conclusions. First, as a more advanced attack method, BBA outperforms TEXTFOOLER and BAE-R that use greedy search during attack generation when using the same substitution generation methods. The substitution generation method of BAE-R also provides more advantages for attacking. Secondly, TEXTGRAD consistently outperforms black-box baselines, indicating the superiority of the gradient-driven optimization method when generating adversarial examples.

Table 8: Performance of proposed TEXTGRAD and baseline methods against normally trained victim models on SST-2 and MNLI-m datasets evaluated by attack success rate. The performance of BBA is evaluated when using substitution generation methods of TEXTFOOLER and BAE-R respectively.

| Attack Method | SST-2 | | MNLI-m | |
|---|---|---|---|---|
| | BERT | RoBERTa | BERT | RoBERTa |
| TEXTFOOLER | 82.84 | 69.38 | 74.82 | 67.33 |
| BBA + TEXTFOOLER | 85.96 | 81.51 | 82.86 | 78.44 |
| BAE-R | 86.62 | 85.92 | 87.00 | 84.56 |
| BBA + BAE-R | 90.80 | 91.2 | 93.61 | 93.77 |
| TEXTGRAD | 93.51 | 96.45 | 94.08 | 95.44 |

# E    PRELIMINARY HUMAN EVALUATION

Besides automatic evaluation, we also conduct human evaluations for justifying the validity of adversarial examples. Here an adversarial example is regarded as valid when its label annotated by a human annotator is consistent with the ground-truth label of the corresponding original example. During the evaluation, each annotator is asked to classify the sentiment labels (positive/negative) of the given sentences, thus requiring no domain knowledge. Note that the ground-truth label is not provided. The following instruction is displayed to the annotator during annotation:

> *Please classify the sentiment of the movie review displayed above. Answer 0 if*
> *you think the sentiment in that movie review is negative and answer 1 if positive.*
> *Your answer (0/1) :*

Our method is compared with BAE, the baseline with the best attack quality according to Table 2 and 3. Specifically, We randomly sample 100 examples from the SST-2 dataset that are successfully attacked by both TEXTGRAD and BAE-R, resulting in 200 adversarial examples in total. These adversarial examples are randomly shuffled and annotated by four annotators. We compute the validity ratios according to the annotations of each annotator as well as the average validity ratio. Finally, the average validity ratios are 43.5% for TEXTGRAD and 39.75% for BAE-R, showing that the quality of adversarial examples generated by TEXTGRAD is slightly better than the baseline. We will also conduct more large-scale human evaluations in the next step.

## F  CONVERGENCE ANALYSIS

In Figure 2, we further show the convergence trajectory of using TEXTGRAD-AT to fine-tune a pre-trained BERT, given by clean accuracy and robust accuracy vs. the training epoch number. We observe that the test-time clean accuracy decreases as the training epoch number increases, implying the accuracy-robustness tradeoff. We also note that the test-time robust accuracy quickly converges in two epochs. This suggests that benefiting from a large-scale pre-trained model, TEXTGRAD-AT can be used to enhance the robustness of downstream tasks in an efficient manner.

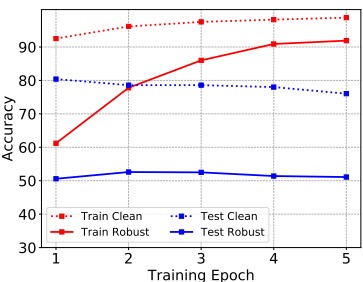

Figure 2: Clean accuracy and robust accuracy on SST-2 train/test dataset during adversarial training.

## G  LIMITATION AND SOCIETAL IMPACT

While the proposed TEXTGRAD can both improve the robustness evaluation and boost the adversarial robustness of NLP models, we acknowledge that there are still some limitations that need to improve in the future. Firstly, TEXTGRAD crafts adversarial examples by synonym substitution. It cannot handle other perturbations such as word insertion, word deletion, and sentence-level modification. We hope to extend our convex relaxation optimization to more perturbations in the future to further promote the performance. Secondly, how to ensemble TEXTGRAD with other types of attacks (for example, black-box baselines) to form a more powerful attack is not explored. Given the success of AutoAttack (Croce & Hein, 2020) in computer vision, it is also plausible to build an ensemble attack in NLP.

With the application of large pre-trained NLP models, the vulnerability of pre-trained NLP models also raises concerns. We admit that TEXTGRAD may be employed to design textual adversarial examples in real life, thus resulting in security concerns and negative outcomes. However, one can still adopt TEXTGRAD for robustness evaluation and adversarial training so as to improve the security and trustworthiness of NLP systems.

