# OpenReview forum: "TextGrad: Advancing Robustness Evaluation in NLP by Gradient-Driven Optimization"
_ICLR.cc/2023/Conference — ICLR 2023 poster_

### Official Review · Reviewer_iGRm · 2022-10-23

**Confidence:** 3
**Clarity, Quality, Novelty And Reproducibility:** 1)	In my understanding, m in Eq (1) i…
**Correctness:** 3
**Technical Novelty And Significance:** 3
**Empirical Novelty And Significance:** 3
**Recommendation:** 6

**Strength And Weaknesses:**

Strengths:
1)	The method is based on an effective convex relaxation method, which is a novel approach to co-optimize the continuously-relaxed site selection and perturbation variables.
2)	According to the results in the paper, the proposed method is able to generate adversarial examples with high quality and high attack success rate in an efficient way.

Weaknesses:
1)	How to maintain a balance between C&W-type attack loss and text fluency regularization?
2)	In Table 5, TEXTGRAD-AT performs much worse than TEXTFOOLER-AT in terms of clean accuracy, while TEXTGRAD has a lower PPL than TEXTFOOLER. I think it is a bit strange why high-quality adversarial examples cannot achieve better clean accuracy.
3)	In Eq (5), how to satisfy the constraints in Eq (2)? For example, 1^T z ≤ k, 1^T u = 1.
4)	The ablation experiments are inadequate and the role of text fluency regularization and C&W-type attack loss was not analyzed.
5)	In Figure 1, why does TEXTGRAD with 1-step attack iteration perform better than the majority of baseline attacks, and is it caused by the preprocessing of substitution words?


**Summary Of The Paper:**

This paper proposes a white-box attack method in NLP that can achieve a high attack success rate and high quality. According to the results in the paper, the proposed method is efficient and effective.

**Summary Of The Review:**

In summary, the proposed method is novel and effective.
On the other hand, I have raised some concerns about the experimental design and results, which may hurt the paper's solidness. The detailed author response has addressed my concerns partly.

---

### Official Review · Reviewer_eerx · 2022-10-26

**Confidence:** 4
**Correctness:** 3
**Technical Novelty And Significance:** 3
**Empirical Novelty And Significance:** 3
**Recommendation:** 6

**Clarity, Quality, Novelty And Reproducibility:**

The paper is generally well-organized and written in general. It is good to see that the problem of text adversarial example generation and its two subproblems (site and substitution selections) can be formulated as a discrete optimization problem. However, a similar idea has been investigated in defense against adversarial attacks, and it would be better to discuss the connection between this work and (Zhou et al., 2021, Dong et al., 2021), and add comparisons.

There are a few typos. For example, \hat{s} in the first line of Equation (1) should be s.

**Strength And Weaknesses:**

Strength:

(1) It is nice to formulate the problems of both the site selection (where to replace tokens) and token modification (which tokens should be selected to replace the original ones) for text adversarial example generation as a discrete optimization problem with convex constrains.

(2) A sampling-based method was also proposed to solve the optimization problem by relaxing discrete constrains into their convex counterparts and using gradient-driven optimization.

(3) They experimentally show that TextGrad achieves improvements in both the attack success rate and the perplexity score over exiting adversarial example generation methods.

Weakness:

(1) A similar idea has been investigated in (Zhou et al., 2021) and (Dong et al., 2021), which relaxes a set of discrete points (a word and its synonyms) to a convex hull spanned, and uses a convex hull formed by a word and its synonyms to capture word substitutions although the goal of their studies is to improve the adversarial robustness of text classification models instead of generating adversarial examples. It would be good to add some text to discuss the similarities and explain the differences.

(2) Although the experimental results show that the TextGrad can be used in adversarial training to improve the robustness of NLP models, the models robustly trained with TextGrad suffer a significant drop in clean accuracy (see table 5).

(3) Many random trials are required to solve the optimization problem via importance sampling, which make the problem method takes about three times the computation time than the greedy-based algorithms.

[1] Zhou, Yi, et al. Defense against synonym substitution-based adversarial attacks via Dirichlet neighborhood ensemble. ACL, 2021.

[2] Dong, Xinshuai, et al. Towards robustness against natural language word substitutions. ICLR, 2021.


**Summary Of The Paper:**

Unlike the computer vision domain, it is hard to generate text adversarial examples efficiently by using the first-order projected gradient descent due to the discrete nature of texts. In this study, a new attack generator, named TextGrad, was proposed to co-optimize the site selection (where to replace tokens) and the substitute choice. To circumvent the difficulty caused by the non-differentiability of discrete optimization, a convex relation method and the corresponding sampling-based solving algorithm has been proposed to map from the continuous optimization space to the discrete token perturbations. The experimental results show that TextGrad achieved some improvements in the attack success rate while the generated adversarial examples have lower perplexity scores.

**Summary Of The Review:**

A text adversarial example generation method, named TextGrad, was proposed in this study, in which the site selection (where to replace tokens) and the substitute choice can be co-optimized in a discrete optimization problem. A sampling-based method was also proposed to solve this discrete optimization problem by relaxing discrete constraints into their convex counterparts and using gradient-driven optimization. However, the experimental results show that the models trained with TextGrad suffer a significant drop in clean accuracy and TextGrad requires about three times the computation time than existing greedy-based methods. The similarities and differences between this work and (Zhou et al., 2021, Dong et al., 2021) also need to be discussed.

---

### Official Review · Reviewer_J6Pr · 2022-10-27

**Confidence:** 3
**Correctness:** 3
**Technical Novelty And Significance:** 3
**Empirical Novelty And Significance:** 3
**Recommendation:** 8

**Clarity, Quality, Novelty And Reproducibility:**

There are challenges make the development of PGD-like NLP attacks difficult. This paper proposes TEXTGRAD, an effective attack method based on gradient-driven optimization in NLP to solve those challenges.

**Strength And Weaknesses:**

Strength:
1. The motivation of this paper is very clear. There are 2 challenges lie for bridging the gap between CV PGD and NLP PGD. (1) the discrete nature of textual inputs together with the strong coupling between the perturbation location and the actual content. (2) the additional constraint that the perturbed text should be fluent and achieve a low perplexity under a language model. These challenges make the development of PGD-like NLP attacks difficult.
2. The proposed method sounds and the paper is in a good shape. This paper develops an effective convex relaxation method to co-optimize the continuously-relaxed site selection and perturbation variables, and leverage an effective sampling method to establish an accurate mapping from the continuous optimization variables to the discrete textual perturbations.
3. Experiment part is solid which shows that the proposed method not only achieves remarkable improvements in robustness evaluation
but also boosts the robustness of NLP models with adversarial training.

**Summary Of The Paper:**

In contrast to the computer vision (CV) domain where the first-order projected gradient descent (PGD) is used as the benchmark approach to generate adversarial examples for robustness evaluation, there lacks a principled first-order gradient-based robustness evaluation framework in NLP. To bridge the gap, this paper proposes TEXTGRAD, a new attack generator using gradient-driven optimization, supporting high-accuracy and high-quality assessment of adversarial robustness in NLP.

**Summary Of The Review:**

The motivation of this paper is very clear. The proposed method sounds and the paper is in a good shape. Experiment part is solid.

---

### Official Review · Reviewer_7WNo · 2022-10-31

**Confidence:** 3
**Correctness:** 3
**Technical Novelty And Significance:** 3
**Empirical Novelty And Significance:** 3
**Recommendation:** 5

**Clarity, Quality, Novelty And Reproducibility:**

The paper is clearly written and easy to follow. The presented idea seems original (but has weaknesses as pointed out above). With the provided details, the results could be replicated.

**Strength And Weaknesses:**

Strengths:
1. The proposed algorithm clearly outperforms baselines on attack success rate as well as perplexity. Ablations show the importance of each setup.
2. The paper is written well and easy to follow in most parts.


Weaknesses:
1. The algorithm seems to be lacking a crucial aspect of preserving the original label. The proposed algorithm replaces a subset of words in the original sentence with some candidates provided by MLMs. This has glaring issues wrt to the definition of an adversarial example. For any position in the sentence, an MLM can easily generate antonyms as candidates which will result in a fluent sentence but flip the label. So it would not really be an adversarial example. The human evaluation does show a slight improvement in label preservation over baselines but it is still low at 43%. Without label preservation guarantees, the evaluation of the attack success rate is meaningless. The evaluation setup which shows: "of the adversarial examples that preserve label, how many led to an attack success" would be more meaningful.
2. The fluency criterion used as part of the loss is conceptually not convincing either. The author chose to use MLM probabilities as a way to evaluate fluency. MLMs by design are not suited to measure the plausibility of full sequences. Since the final evaluation is done by perplexity, a fluency criterion defined by an autogressive (aka causal) LM would be better suited as an objective. Additionally, human evaluation should also involve evaluating fluency.

**Summary Of The Paper:**

This paper presents a method of white-box adversarial attacks against text classification systems. Given a classification model, the algorithm involves taking an input example and transforming it to construct an adversarial example by (1) selecting a subset of positions to replace, (2) for each selection position, selecting a word from a given set of words for that position while constraining these selections to be lead to a fluent sentence. This work executed this idea by formulating this search problem as continuous optimization with the primary goal to flip the model label for x with a constraint on fluency of the output. On several text classification tasks with 3 different models, the authors report better performance in terms of generating sentences that are more successful at flipping the label compared to that of original x and have lower perplexity. On a small scale human evaluation, the generated adversarial samples seem slightly better than baselines at preserving the original sentiment.

**Summary Of The Review:**

The presented algorithm has interesting ideas for generating adversarial examples for text classification systems and shows improvement on attack metrics, however one major criterion for evaluating the adversarial examples is label preservation on which the authors provide a very small scale analysis which is not entirely convincing of the utility of the method. Finally, I would add that I am not an expert in this space and provided a rebuttal, willing to revise my score.

---

### Decision · Program_Chairs · 2023-01-20

**Decision:**

Accept: poster

**Justification For Why Not Higher Score:**

The paper has several issues as pointed out in the weakness. Most of these key issues (e.g., label preserving concern) are common for papers published in this line of research, so it might not be fair to blame this paper. However, on the other hand, keeping accepting papers with these issues may make this research line stuck at a local minimum and hinder the researchers from focusing on the real problem.

Overall, the paper is on the borderline between accept (poster) and reject. Weighted between the strengths and weaknesses, I suggested the paper be accepted, but the decision can be bumped down.

**Justification For Why Not Lower Score:**

The paper is a decent work with quality aligned with previous work published on this topic in venues similar to ICLR.

**Metareview: Summary, Strengths And Weaknesses:**

The paper studies using first-order projected gradient descent for generating adversarial attacks for text classifiers.

Strengthens:

+ The research and proposed methods are well-motivated and the paper is easy to follow in general.

+ The proposed approach based on the convex relaxation method is interesting and novel. Although some earlier works resemble the proposed approach as pointed out by the reviewers, there is a sufficient difference between this and prior work. However, the paper should position itself with the literature better compared with prior work.

+ The experiments are sufficient to support the main claims of the paper. There is no major concern after the rebuttal.

Weaknesses:

- The label-preserving problem pointed out by 7WNo is critical. Without preserving the labels, the definition of adversarial attacks breaks and invalidates the experiment results. I understand several recent papers omit this issue and still get published; however, in a long run, it is essential to ensure the label-preserving property is verified. Although the authors provide additional arguments and promise to do the human evaluation in the rebuttal, the manuscript hasn't been updated. I would highly recommend the authors address this issue in the revision.

- Perplexity score from MLM score does not always align with fluency. The use of terminology should be more rigorous to avoid confusion.

- More discussion about the tradeoff between clean accuracy and robust accuracy should be provided in the revision.

- All the experiments of this paper are on text classification problems. I would suggest the paper changes the title to emphasize the focus is on "Robustness Evaluation in Text Classification". NLP consists of more complex language problems like QA, translation, parsing, etc. The proposed approach cannot deal with those tasks.

Missing reference:

Using projected gradient to generate adversarial attacks for text classifiers is discussed in the following paper. Although the approach is different, it should be discussed:
Adversarial Training with Fast Gradient Projection Method against Synonym Substitution Based Text Attacks, AAAI 2021.



**Note From Pc:**

if the above contains the word "oral" or "spotlight" please see: "oral" presentation means -> notable-top-5% and "spotlight" means -> notable-top-25%. As stated in our emails, we are disassociating presentation type from AC recommendations

**Summary Of Ac-Reviewer Meeting:**

N/A